# Increasing Access to Healthy Foods through Improving Food Environment: A Review of Mixed Methods Intervention Studies with Residents of Low-Income Communities

**DOI:** 10.3390/nu14112278

**Published:** 2022-05-29

**Authors:** Dea Ziso, Ock K. Chun, Michael J. Puglisi

**Affiliations:** Department of Nutritional Sciences, University of Connecticut, Storrs, CT 06269, USA; dea.ziso@uconn.edu (D.Z.); ock.chun@uconn.edu (O.K.C.)

**Keywords:** food insecurity, chronic disease, multilevel approaches, low-income, food environment

## Abstract

Food insecurity is a broad and serious public health issue in the United States, where many people are reporting lack of access to healthy foods. The reduced availability of healthy, affordable foods has led to increased consumption of energy-dense and nutrient-poor foods, resulting in increasing the risk for many chronic diseases such as obesity, cardiovascular diseases, and type 2 diabetes mellitus. Thus, identifying promising approaches to increase access to healthy foods through improving the food environment is of importance. The purpose of this review article is to highlight how the food environment affects directly a person’s food choices, and how to increase access to healthy foods through improving environmental approaches. The literature search was focused on finding different approaches to improve food security, primarily those with an impact on food environment. Overall, potential solutions were gathered through multilevel environmental approaches, including nutrition education and peer education, community-based participatory research, and policy changes in supplemental nutrition programs. A recommendation to reduce food insecurity is learning to create meals with a variety of seasonal fruits and vegetables purchased from affordable farmers’ markets.

## 1. Introduction

Food insecurity comprises limited or uncertain access to nutritious and adequate food intake and is widespread in the United States [1]. The United States Department of Agriculture (USDA) reported that 10.5% of Americans were food insecure at least some of the time during 2019 [2]. Many individuals who are food insecure utilize food banks and food pantries to procure food, but studies have shown that pantry users consume limited amounts of fruits, vegetables, and fiber [3]. The majority of the calories for this population are taken from energy-dense, nutrient-poor foods, including refined grains and foods high in added sugars and saturated fats, in contrast with the food-secure population, who have access to more nutrient-dense and healthier options [4]. This can result in the consumption of empty calories, rather than insufficient calories [4]. In general, food-secure individuals are more likely to meet the recommended dietary allowance for nutrients than food-insecure individuals [5]. This is displayed in a study by Champagne et al. [5], in which researchers determined the food security status through self-reported food intake, and found that food insecurity is associated with lower-quality diets assessed by the healthy eating index (HEI) scores.

A few factors that affect poor diet quality are related to lack of access to healthy foods in the surrounding neighborhood and limited household income [3,6]. The macro-environmental sector refers to broad infrastructure (including food advertising and health systems), whereas micro-environments indicate local settings (such as workplaces and homes) [7]. Through improving these food environments, access to healthy food options can be increased, resulting in better diet quality and reduced chronic disease risk in these populations. Therefore, developing effective strategies to improve the diet quality and nutritional status of high-risk populations is important for the prevention of diet-related chronic disease. This review paper will evaluate specific risks for chronic disease associated with food insecurity and strategies to improve the food environment and individuals’ choices towards a healthier diet and lifestyle. The objectives will be reached through assessing the literature on mixed-method intervention studies with residents of low-income communities.

## 2. Materials and Methods

To learn more about food insecurity and the risk of chronic disease, as well as approaches to improve food access, a literature search was conducted in June 2021 using the databases PubMed, Scopus, and Google Scholar, with a combination of keywords “food insecurity”, “chronic disease”, “food access”, “multilevel approaches”, “low-income”, “healthy eating”, “community”, and “policy changes”. Reading through each study, the titles and abstracts were observed to identify which articles provided informative data that helped towards reducing food insecurity. As summarized in Table 1, full-length articles and book chapters in English with the US population in focus for all age ranges were selected. There were no limits to the state or region in the US in which the studies were conducted. The study designs conducted were randomized controlled trials, clinical trials, comparative studies, multicenter studies, and cohort studies. Notes, comments, editorials, and review articles were excluded from the articles with an exception of two review articles that were complementary to the data from other major studies. Studies that had other populations in focus, including low-income communities outside of the US, were excluded. Additionally, there were studies selected from the reference lists of relevant articles. Studies that compared the lifestyle and diet of food-secure populations vs. food-insecure populations were included, with a specific focus on the impact of the food environment. As different approaches to improve food security were searched for this narrative review, there was a focus on nutrition and peer education, community-based research, policy changes, and multilevel approaches.

##  3. Food Insecurity and Risk for Diet-Related Chronic Disease

### 3.1. Obesity

At the individual level, rates of obesity are higher among groups with low education and low incomes [8,9]. Current meta-analysis studies show that food insecurity increases the risk for obesity for adults (odds ratio 1.15) in food-insecure households, especially women odds ratio of 1.26) [10]. Paeratakul et al. [9] assessed data from the USDA Agricultural Research Service Continuing Survey of Food Intakes by Individuals, and reported that, among other factors, socioeconomic condition is one of the main factors leading to greater obesity severity. The study further linked obesity rates with an increased incidence of diabetes, hypertension, and high serum cholesterol, which further supports the role of socioeconomic factors in increasing these disease risks through an increase in obesity [9]. Individuals with food insecurity commonly use resources, such as food pantries, for access to a variety of products. An important issue that has been shown to lead to obesity among low-income populations is the poor nutritional quality of food and lack of knowledge of how to prepare certain fresh produce provided by the food pantries and other food assistance programs [5,11]. A great focus has been towards children in low-income populations and their greater risk of obesity due to their diet patterns and food choices [12]. Kaur et al. [12] analyzed the National Health and Nutrition Examination Survey (NHANES) data assessing personal food insecurity through USDA’s Food Security Survey Module to determine its relationship with obesity risk. The researchers determined that obesity was significantly associated with levels of food insecurity among children of ages 6 to 11 years, with an odds ratio of 1.81; 95% confidence interval (CI) 1.33 to 2.48 [12].

Food pantries are good resources from which to analyze the needs of food-insecure populations and determine obesity rates. Studies of food pantry participants concluded that the mean body mass index (BMI) of the pantry users was 29.5 kg/m^2^, and 78.0% of the population of obese pantry users were women [3,5,11,13]. Many people from the population have shown an interest in regularly consuming nutritious food and fresh produce, but they reported that these products were unaffordable [14,15].

### 3.2. Cardiovascular Diseases

According to the American Heart Association, the seven cardiovascular health metrics are based on smoking, diet, physical activity, BMI, blood pressure, total cholesterol, and fasting glucose [16]. Individuals with food insecurity face barriers meeting the ideal cardiovascular health metrics, including lower odds of meeting at least three metrics (*p* < 0.01) [17]. Data further show a greater presence of hypertension and heart disease in individuals with lower education and income compared to others [9]. Furthermore, hypertension is associated with greater intake of added sugar and sugar-sweetened beverages (*p* < 0.05) [18]. Adults from food-insecure households had a 21% higher risk of clinical hypertension than adults from food-secure households [19]. A systemic review and meta-analysis of 36 studies indicated an association between food-insecure adults and self-reported hypertension with an odds ratio of 1.13 [20]. Additionally, very low food security was associated with increased risk for cardiovascular disease, and a 58% higher risk for cardiovascular disease mortality [21,22,23]. Sun et al. focused on associations of adult food insecurity with all-cause and cardiovascular disease mortality in US adults [23]. This study concluded that participants with very low food security had a higher risk of cardiovascular disease compared with those with high food security, with an odds ratio of 1.54 (95% CI 1.04–2.26) [23].

Bazzano et al. [24] conducted the first NHANES Follow-up Study to observe fruit and vegetable intake through a food-frequency questionnaire, and the incidence of mortality from cardiovascular disease from medical records and death certificates. A significant association was identified between frequency of fruit and vegetable intake and incidence of and mortality from stroke, ischemic heart disease, and cardiovascular disease [24]. Bazzano et al. [24] further concluded that consuming three or more fruits and vegetables per day, compared to less than one per day, was associated with a 27% lower stroke incidence, a 42% lower stroke mortality, a 24% lower ischemic heart disease mortality, a 27% lower cardiovascular disease mortality, and a 15% lower all-cause mortality. Due to the high cost of fruits and vegetables, lack of transportation, low quality products in low-income areas, and other factors, individuals in these neighborhoods consume fewer servings of fruits and vegetables than the Dietary Guidelines recommendations, leading to an increased risk for cardiovascular disease [25]. Among nonelderly adults with household income <200% of the federal poverty level, analysis of NHANES data by Seligman et al. [19] (a nationally representative survey of the US population) found an association between food insecurity and clinical evidence of hypertension and diabetes. 

### 3.3. Diabetes Mellitus

As previously mentioned, poor diet quality includes low nutrient consumption and high intake of energy dense foods. This lifestyle factor is one of the main factors leading to diabetes mellitus. Walker et al. [13] conducted a cross-sectional analysis with data from the NHANES survey 2005–2014 to evaluate levels of food security and its association to diabetes. The results indicated higher risk for food insecurity, with an odds ratio for prediabetes of 1.39 (95% CI 1.21–1.59), for undiagnosed diabetes of 1.81 (95% CI 1.37–2.38), and for diagnosed diabetes of 1.58 (95% CI 1.29–1.93) [13]. The results are supported by a cross-sectional study; when compared to individuals with food security, participants with low food security were 1.35 times more likely to have prediabetes (95% CI: 1.17–1.55) [26]. Very-low-food-secure participants, compared with both low-food-secure and food-secure participants, have been reported to have greater diabetes distress and more frequent and severe hypoglycemic episodes [27]. Individuals with food insecurity live paycheck by paycheck or wait for monthly assistance, which can lead to a cycle of restraining dietary intake during hard times and overeating during food restock [28]. This process can lead to insulin resistance and diabetes [13,28,29]. In individuals with food security the self-management to improve diabetes is easier, but a lack of quality food makes diabetes self-management more difficult, worsening the food-insecure individual’s health condition [27]. A meta-analysis study by Abdurahman et al. further strengthens the hypothesis for an association between household food insecurity and increased risk of type 2 diabetes, with an odds ratio value of 1.27 (95% CI 1.11–1.42) [30]. Based on the studies mentioned previously, it appears that food insecurity is associated with different stages of diabetes mellitus; however, there are some recent studies that contrast the findings and do not suggest an association between food insecurity and clinically determined type 2 diabetes or significant differences in fasting blood glucose and HbA1c [31]. Beltran et al. additionally stated that food insecurity is a complex issue and it might look different depending on factors such as social, economic, and geographic consideration [31]. The study explained that the differences in the findings might be a result of the difference in food insecurity concepts, where in some cases hunger and chronic starvation are the primary drives of food insecurity [31]. In another study, starvation due to food insecurity was related to worsening signs of insulin sensitivity in type 2 diabetes [29]. Meanwhile, food-insecure areas in the United States are usually not associated with hunger, but mainly with intake of lower-quality, high-calorie foods, which increase a person’s risk for type 2 diabetes [31].

## 4. The Impact of Food Environment on Access to Healthy Food Choices

The nutritional environment is affected not only by the number of stores in an area but also the availability and cost of healthy food items. Reduced access to fresh produce greatly impacts food choices for low-income populations [32]. Typically, individuals with reliable transportation make frequent trips to supermarkets [32,33]. However, many people in low-income neighborhoods rely greatly on food pantries due to the lack of transportation and availability of fresh produce in their areas [33]. A research study, conducted in a low-income neighborhood in Pomona, CA, determined that 41% of the food pantry clients did not live within walking distance of a store with a variety of fresh produce and 13% did not have access to any type of food store with fresh produce [33]. This problem is widespread in many areas in the US and highlights rural adults not being able to meet recommended nutrition guidelines due to environmental barriers and lack of community resources [33]. In areas with food insecurity, convenience stores are more common than supermarkets or grocery stores [32]. These stores have a very limited range of food items and are more likely to stock less healthy versions of products (for example: low-fiber bread vs. high-fiber bread, whole milk vs. reduced-fat milk vs. low-fat or nonfat milk) [32]. A multicenter study in Hartford, CT, combined customer shopping behavior with store food inventory data, and concluded that when there is a greater variety of fresh products, such as fruits and vegetables, there is an increased likelihood of these products being purchased [34]. This research study conducted face-to-face interviews on different days and at different times of the day to measure typical food shopping behaviors and determine whether shoppers had access to and used the Supplemental Nutrition Assistance Program (SNAP) and Special Supplemental Nutrition Program for Women, Infants, and Children (WIC) benefits in the store [34]. They also maintained inventories, using a modified version of the Nutrition Environment Measures Survey in Stores (which included the availability of fresh/canned fruits and vegetables, and whole grain and reduced-fat dairy products) to measure availability and quality of healthy food in stores [34]. Due to an increased variety of fruits and vegetables offered, those receiving SNAP were significantly more likely to purchase fruit (*p* < 0.05) and vegetables (*p* < 0.01), compared with those who were not receiving SNAP [34].

When it comes to the cost of fresh products, more healthful versions of food items are typically more expensive than the corresponding less-healthful versions, with the exception of milk [35]. At the same time, foods with high energy density provide the most calories at a lower cost, which contributes to people in low-income populations consuming them over fresh, nutrient-dense products [35]. Alteration of this environment, as with the study discussed above by Martin et al. [34], may significantly change these consumption habits.

## 5. Environmental Approaches to Improve Food Security and Nutritional Status: Multilevel Intervention Studies with Residents of Low-Income Communities

Figure 1 uses the framework of the socioecological model (SEM) concept, as outlined by the Centers for Disease Control and Prevention [36], to summarize the key multilevel approaches discussed in this review for improving food security and nutritional status. Environmental approaches and nutrition programs can improve food security more efficiently if they work in a multilevel interaction approach [36]. This interaction starts from working personally with the individual, to creating a support system of family, friends, and social networks [36]. Other more significant factors that can assist this population are resources provided through schools or workplaces, communities (considering cultural values and norms), and policy changes from local laws to national changes [36]. Multilevel approaches have been used to increase the fruit and vegetable intake in low-income housing communities. Table 2 summarizes the key approaches that have led to improvements in food security. A 12-month cluster, randomized controlled trial (RCT) was conducted to demonstrate the benefit of year-round fruit and vegetable markets at improving fruit and vegetable (F&V) intake for low-income adults [37]. This research study featured discount fresh fruit and vegetable markets called ‘Fresh to You’ (FTY), as well as a multicomponent, educational intervention, which was a combination of individual, community, and policy change approaches [37]. The prices of the fruits and vegetables were kept at or below the retail prices at local supermarkets [37]. This study also included chef-run cooking demonstrations, taste-testing events, shared recipes, and educational boxes (including newsletters, DVDs, reusable shopping bags, and kitchenware) and two six-week educational/motivational campaigns, which focused on increasing intake and variety of fruits and vegetables [37]. The results indicated that more than half of the participants attended a few of the FTY markets [37]. From baseline to 12 months, there was an increase in total fruit and vegetable intake by 0.44 cup/day (as assessed by the National Cancer Institute’s ‘Eating at America’s Table All Day Screener’) in the intervention group with the control group decreasing intake by 0.08 cup/day [37]. There was also an increase in the frequency of fruits and vegetables by 0.24 servings per day, which was assessed through the Fruit and Vegetable Habits Questionnaire Score [37]. Reading newsletters or attending campaign events was not associated with any change in fruit and vegetable consumption, but watching DVDs was associated with an increase in fruit and vegetable intake by 0.69 cup/day [37].

Adding culturally competent aspects to a community approach, with resources focused on multiethnic communities, materials provided in the population’s native language, and incorporation of recipes and food demonstrations from their country of origin, may result in improvements in diet quality [38]. Hammons et al. [38] added these cultural aspects and reported greater interest for the individuals to try recipes provided with fruits and vegetables and an overall increase in daily servings of fruits and vegetables consumed [38]. The results are supported by another study that reported an increase in variety of fruits and vegetables (*p* < 0.01 for both), after a six-week cooking program at the Community Food Bank that teaches food pantry clients plant-based recipes [39]. These data further indicated a decrease in purchases of carbonated beverages and desserts with a value of *p* < 0.01 [39].

B’more Healthy Communities for Kids was a multilevel obesity prevention study that sought to improve healthier food purchasing and help reduce sweet-snack consumption among low-income African-American youths [40]. This group worked to achieve the objectives for two-year time period, through youth-led nutrition education in recreation centers, in-store promotion, and social media programs [40]. In this multilevel approach, many areas for education were considered, including promoting healthier alternatives to beverages and snacks, and sharing information for healthier cooking methods through promoting healthier cooking ingredients, such as low-fat milk, 100% whole wheat bread, and fresh/canned/frozen vegetables [40]. The multiple components involved wholesales, which were encouraged to stock B’more Healthy Communities for Kids-promoted healthier food alternatives [40]. They also worked on improving supply for healthier options of foods and beverages in corner stores and carryout restaurants, as well as improving demand through taste tests for healthier alternatives [40]. Posters, handouts, and educational sessions were provided. Through these sessions there were peer-led, hands-on activities where participants learned different sugar and fat contents in drinks and snacks and were introduced to the traffic light labeling method for beverages and snacks [40]. For additional resources, recipes, news, and educational activities related to healthier eating behaviors were sent through social media and texting [40]. The text message platform was focused towards caregivers with goal setting strategies and educational activities, where they received messages three to five times a week related to healthier eating behavior [40].

B’more Healthy Communities for Kids also worked with key city stakeholders to support policies for a healthier food environment and provided evidence-based information to support the development of policies at the city level using a Geographic Information System/System Science simulation model to help stakeholder decision-making in regard to sugar-sweetened beverage warning labels, urban farm tax credits, mobile meals, etc. [40]. The results showed that the overall intervention group purchased 1.4 more healthier foods and beverages per week in relation to the comparison youth. Additionally, there was a decrease in calorie intake from sweet snacks and desserts among older intervention youths [40]. This finding was supported by other results in improving the availability of healthier foods and beverages in small food stores in intervention zones, indicating that food availability affects an individual’s choice [40]. Overweight and obesity are major issues, especially in low-income communities, but community-enhanced school programs can be effective in reducing childhood obesity in these populations [41]. Schools with health program training and community partnerships decreased the percentage of students classified as overweight/obese by 8.3%, compared to a 1.3% decrease in schools that were provided only with health program training, without the community aspect [41].

In the case of emergency food aid, there are also food banks open and available around the US. Wetherill et al. [42] conducted a study to look a strategies and innovative programs that are focused on advancing nutrition-focused food banking in the United States. This study included in-person or phone interviews to obtain further information regarding personal experiences, perceptions, and practices related to nutrition-focused food banking [42]. Overall, the study findings indicated that food banks are implementing a variety of multi-level approaches to improve healthy food access among users [42]. This is done through four major themes: building a healthier food inventory at the food bank; enhancing partner agency healthy food access, storage, and distribution capacity; nutrition education outreach; and expanding community partnerships and intervention settings for healthy food distribution, including healthcare and schools [42].

### 5.1. Individual Nutrition Education and Peer Education

The main factors that have been shown to increase fruit and vegetable consumption in multilevel approaches are taste tests and nutrition education, including providing healthy recipes for participants [37]. Along with healthy recipes, some studies have focused on helping improve participants’ cooking skills to make low-cost meals, which further decreased food insecurity, by increasing cooking at home and reducing the amount of time people ate out at restaurants, leading to increased total variety of vegetables and fruits in the diet [34,37,39,40]. Even short-term nutrition education has made people more aware concerning sodium in processed foods and nutrient and calorie-dense foods, the importance of physical activity, and interpreting nutrition levels [43].

WIC focuses on improving the health and nutritional status of pregnant and postpartum low-income women and children up to the age of 5 years. The participants of these programs are faced with many barriers when shopping for the correct foods on their WIC food list, therefore nutrition education is a great tool to better assist this population [44]. Through mobile phone apps, there are educational tools to help the participants with shopping management features, clinic appointments, and informational resources [44]. Shopping management features assisted the WIC participants with real-time shopping for WIC foods in the grocery store, including food benefit balance checking, barcode scanning, and checking if the item was WIC eligible [44]. The participants shared that this app was very easy to use and convenient in helping pick the right products, while saving time [44]. Some applications also allowed participants to manage their WIC clinic visits, and view their future appointments and the documents needed to be provided in the appointments [44]. Required nutrition education modules were also features, which could be completed from home instead of going to the clinic [44]. Throughout a 7 month period, Weber et al. [44] reviewed the main features of a publicly available mobile phone app for WIC participants and concluded that, even though all the features were beneficial and useful, the most frequently used were the shopping management features. Some users shared their feedback, giving the app 4–5 stars out of 5 stars for being time saving, convenient, and an overall great app [44]. Other users would have liked in addition a nutrition/healthy section with recipes and ideas to help people learn to consume healthier choices, not just the WIC-approved foods [44].

Another approach to increase diet quality for low-income populations is individual tailoring of nutrition education [18]. A recent pilot 8 week study assessed the effectiveness and acceptability of personalized nutrition intervention for mobile food pantry users [45]. When comparing the treatment group with the control group, a personalized nutrition education intervention was effective in improving the diet for food-insecure participants (4.54% vs. 0.18% improvement in healthy eating index scores) [45]. Culturally tailored nutrition education involving family time and physical activity has also been a way to incorporate healthier food choices [46,47]. These approaches also include educational information in the participant’s native language, including all the handouts, recipes, and visual guides [47]. Focusing on diet based on culture, studies have also worked with tiendas, small Latino stores, to promote greater intake of fresh produce among consumers [46].

Education programs culturally tailored to a specific group have also been shown to be effective. Flores-Luevano et al. [48] conducted a bilingual culturally tailored, hands-on diabetes education program among Mexican American adults with diabetes. The sessions were interactive with demonstrations, activities to promote problem-solving, and facilitated group dynamics through sharing personal experiences [48]. This, intertwined with peer-education, resulted in improvements in glycated hemoglobin by −1.1% and total cholesterol with −17.2 mg/dL at 6 months post-intervention [48]. There were also behavioral changes, such as glucose self-monitoring improvement by 1.3 times increase a week, increased exercise levels, and increased positive nutritional behavior by 2.23, and the benefits were observed with attendance rates as low as 50% [48].

Marshall et al. [49] conducted a two-year follow-up study using a one-group pre-post evaluation design that focused on school-based nutrition education and food co-op intervention and how it can increase children’s intake of fruits and vegetables. In this study, 407 families completed baseline data, of which 262 parent-dyads agreed to participate in the two-year follow-up study, where the parents were provided with education along with their children [49]. This nutrition education included changes in home setting, such as increased frequency of cooking behaviors, increased usage of nutrition facts labels in making grocery purchasing decisions, and increased food availability of fruits and vegetables [49]. The results of the study showed an increase in child intake of fruits by 0.18 cup/day, vegetables by 0.14 cup/day, and fiber by 1.06 g/1000 kcal, and a significant decrease in total fat intake by 1.55 g/1000 kcal and percent daily calories from sugary beverages by 0.52% [49]. Parents also reported an increase in daily intake of vegetables by 0.6 cups/d and combined fruits and vegetables (*p* < 0.05) [49]. 

### 5.2. Community-Based Participatory Research

Community-based participatory research is a kind of study where community members, organizational representatives, and academic researchers are all equally involved in the process [50]. This research method is important to collect the data on what kind of lessons and resources will be beneficial for the population, but also to make the participants more comfortable and more open when conducting interviews with educators of the community. This approach was used in the selected studies to modify elements in the environment, which would result in an increase in nutrient-dense food consumption. A number of studies have focused on improving diet quality and reducing metabolic risk through gardening and cooking [47,48,51]. These studies organized classes tailored towards low-income youths and consisted of lessons about gardening, where they used “hands-on” approaches to facilitate participation in planting, growing, and harvesting organic fruits and vegetables [47,48,51]. Additional interactive cooking and nutrition lessons were included with the fruits and vegetables raised from the garden [47,51].

These elements worked to increase fruit and vegetable consumption and preparation of healthy snacks and meals [47,51]. Creative food preparation with blending of new vegetables into juices or other dishes has also been a way to introduce unfamiliar foods to children and other family members [47,51,52]. Participants shared that their children would be curious of the new items introduced each week and they were excited to include them in their smoothies [47,51,52].

As previously mentioned, accessibility and affordability are two of the main factors that lead to food security. Since supermarkets and convenience stores are known to have expensive products, a way to provide access to food in low-income families is community-supported agriculture products. Community-supported agriculture products are more affordable and flexible in their accessibility [52]. Farm Fresh Foods for Healthy Kids examined the perception of food access among low-income families in nine communities participating in community-supported agriculture [52]. Participants reported improved access to food products and benefited from flexible pick-up times and locations; however, despite the cost being relatively low, payment remained a barrier for some [52]. A multistate randomized intervention trial targeted obesity prevention in low-income families through improving access to affordable, local, seasonal produce through community-supported agriculture and support of obesity-related behavior changes through tailored education to increase knowledge and skills, and provide increased revenue and business to support community-supported agriculture farmers [53]. Even though the community took steps to help low-income populations, there are still barriers that need to be faced for more effective results of current and future studies [53]. When shopping through community-supported agriculture, participants believed that they were saving money for produce of high quality, compared to the grocery stores [53].

McGuirt et al. [54] examined the influence of farmers’ market pricing and accessibility on willingness to shop at farmers’ markets, among low-income women. Percentage price savings were presented visually as discounts at the standard amount, or there were pictures of the amount of produce a consumer could buy at the farmers’ market compared to the supermarket, reflecting the savings [54]. The different quantity bought with the same price was determined by a member of the research group who went to local supermarkets to establish the price per pound and calculate the amount to compare with the products from the farmers’ market [54]. The results of this study showed that there was an increased interest to shop at farmers’ markets when there was at least a 20% price saving [54,55]. Additionally, participants were more influenced by the visual representation of a greater quantity of produce with the price savings, rather than the money saved by the reduced price [54].

**Table 2 nutrients-14-02278-t002:** Summary of key studies to increase access to healthy foods.

Author	Type of Study	Target Population	Sample Size	Type of Approach	Outcome Measure	Results
Gans. et al. [37]	RCT	Western adults	1587	Individual, community, policy Changes	Fruit and vegetable consumption measured by National Cancer Institute’s “Eating at America’s Table All Day Screener”	-↑ total intake F & V by 0.44 c/day with the control group ↓ by 0.08 c/day (*p* < 0.02). -↑ F&V frequency (*p* = 0.01)
Trude. et al. [40]	RCT	Obese children (9–15 years old) in 30 areas of Baltimore.	401	Individual, interpersonal, organizational, community, policy	-Purchase and consumption of low-sugar foods and beverages.	-↑ healthier purchases by 1.4 more items per week compared to the control group. -There was a 3.5% ↓ in kcal from sweets for older intervention youths.
Weber. et al. [44]	Review and analysis of features	WIC participants	17 app features	Organizational and community	-Reviewing app stores and their benefits to users.	App features were classified into categories for shopping management, WIC required nutrition education modules and others. The app was rated with 4–5/5 stars
White et al. [52]	Multicenter randomized intervention trial	Children	53	Community and policy	-Increasing food access based on availability, accessibility, affordability, acceptability, and accommodation.	Availability was enhanced for those who could select their own produce items. Flexible pick-up times and locations. ↑ access to F&V.
McGuirt et al. [54]	Qualitative Study	Women of child-bearing age	37	Individual, organizational, and policy	-Examine willingness to shop at farmers’ markets.	More likely to shop at farmers’ market when price saving ↑ at least 20%.

↑ indicates increase, ↓ indicates reduction.

### 5.3. Policy Changes in Supplemental Nutrition Programs

To reduce energy-dense, low-nutrient food consumption, there need to be environmental and policy changes that promote healthy eating. The main categories that need to be in focus, based on research, are pricing, nutrition labeling, and access to healthier ready-to-go foods [55,56,57]. Additionally, to increase food security and diet quality, policy changes are needed in school programs that work on strengthening links to the traditional, nutrient-dense food system in schools [58]. A significant increase in fruit and vegetable intake has been shown with greater access to healthier ready-to-go foods [58]. Additionally, reducing the cost of healthier snacks increases the consumption of these products [57,59]. When prices of fruits and vegetables were reduced by 50% for high school students, their consumption increased by twofold to fourfold [55]. Low-fat snack sales increased by 93% with a 50% reduction in their prices [59]. Studies also worked with key city stakeholders to support policies for a healthier food environment and provided evidence-based information to support the development of these policies [40].

## 6. Discussion and Conclusions

Food insecurity is a widespread problem in the US that greatly affects quality of life, leading to greater risk for obesity, diabetes, and cardiovascular diseases. There are a number of barriers causing food insecurity in certain areas, such as lack of transportation, food deserts or food swamps, and lack of nutrition education. This review paper discussed certain approaches to reduce food insecurity in low-income communities and increase access to healthful foods, especially consumption of fruits and vegetables. Multilevel approaches are shown to have the most distinguishable results and also take into consideration a wide spectrum of reasons and factors. Multilevel approaches have included nutrition educational material, taste-testing events, price reduction of healthy products, increased access to healthy options, and overall policy changes. The major limitation to multilevel approaches is that when a change is observed, there is not a way to specifically identify which components of the intervention led to changes in food behaviors [35]. Other potential limitations of studies thus far have included small sample sizes and sample populations that were predominantly women, self-reporting of data, potentially leading to bias, and the possibility that participants may not have fully represented the low-income population of interest [38,39,53,55]. Beneficial approaches comprise community-based research, which obtains an input from the community, where the main focus is on factors that lead to food insecurity and how to reduce these barriers. Farmers’ markets that provide local, seasonal, affordable produce are shown to be a way to support behavior changes and increase access to fresh produce [12]. Nutrition education helps decrease food insecurity from a different approach, through hands-on activities and peer-education to increase cooking skills and help incorporate a variety of fruits and vegetables in the diet [17,37,49].

Additionally, current studies have shown a need for more research, but it is important to point out that perception of an individual’s food environment may impact the foods they purchase and consume [60]. For example, if they perceive the environment to be poor, they may be less likely to buy fruits and vegetables and other healthy options [60]. Furthermore, future research can further focus on how to effectively improve diet quality and reduce diet-related chronic disease risk by developing and validating multi-dimensional intervention studies tailored for target populations with special needs and barriers, and studying the impact of the perceived food environment and social support on improving the diet quality of a population with poor access to healthy foods.

## Figures and Tables

**Figure 1 nutrients-14-02278-f001:**
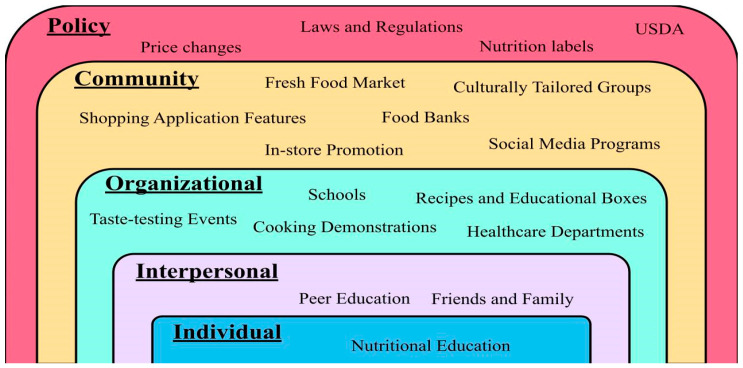
Multilevel approaches to increase healthy food consumption in low-income populations, based on the SEM [36].

**Table 1 nutrients-14-02278-t001:** Criteria for studies included.

Parameter	Criteria	Exclusion
Search terms included	Food insecurity, chronic disease, multilevel approaches, low-income, food environment.	N/A
Criteria for study design	RCT, clinical trials, comparative studies, multicenter studies, cohort studies, qualitative studies, books.	Review articles, notes, comments, editorials.
Criteria for subject population	Low-income populations, all age ranges.	Other populations, low-income population not living in US.
Criteria for outcomes	Improved dietary behavior, weight status, improved lifestyle, educating the community on healthier choices when on a budget.	N/A

## Data Availability

Not applicable.

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
