# Peer review of "Increasing Access to Healthy Foods through Improving Food Environment: A Review of Mixed Methods Intervention Studies with Residents of Low-Income Communities"

_nutrients, 2022, doi:10.3390/nu14112278_

Round 1

Reviewer 1 Report

The authors prepared an overview about improving food environment and its role in access to healthy foods in food insecure populations in the USA. It is an interesting, hot topic, but the article in its current form does not seem to be too up to date for readers due to the paucity of current literature (literature from the last two years) and the lack of meta-analyses with the highest level of evidence.

Therefore, during the review process some major and minor comments, questions were raised. Although it is not a systematic review, the authors should at least indicate when the literature was searched. Since the most recent article is from 2020 December, I assume that it was done more than a year ago (early 2021) and that no recent literature search has been done since then. Please clarify it.

You should also clarify the population with age range. Have you studied the effects of food insecurity only in adults, or also in adolescents and children? It should be included in Table 1, Criteria for subject population.

In the recent 2-3 years many systematic reviews and meta-analyses have been published on food insecurity and risk of chronic diseases (obesity, T2DM, CVD), mainly based on data collected in the US population, however I couldn’t find either these meta-analyses or these more recent US (original) publications. Writing on the basis of a single study that food insecurity (FI) predisposes to obesity is less convincing than writing on the basis of a meta-analysis. I know that these meta-analyses use data outside the US also but you can start a statement based on a meta-analysis and then write that US data show also in the same direction. Meta-analysis about FI and obesity: Moradi et al, 2019 (https://pubmed.ncbi.nlm.nih.gov/30219965/), FI and hypertension: Beltrán et al, 2020 (https://pubmed.ncbi.nlm.nih.gov/33201873/), FI and T2DM: Beltrán et al, 2022 (https://pubmed.ncbi.nlm.nih.gov/34726354/).

In the T2DM part you cite Seligman et al, 2007 based on data collected 1999-2002, however, more recent data of the same population has also been published (Walker et al, 2018: NHANES 2005-2014). If there is available, why don’t you use the more recent data? This would also confirm the relevance of your publication.

In part 5 you discuss multilevel intervention studies performed in the US. In rows 198-203 you show results of Hammons et al, however, this publication cannot be found in Table 2. What determined whether a publication was included in this Table or not? This inconsistency is very confusing. All relevant publications should be included.

I have more concerns about Table 2. It describes multilevel approach studies, however, it is not obvious why are they multilevel? You should include these multiple levels. I don’t think so that Table 2 should contain Title of the articles, this column can be deleted; readers can find these articles in the References based on the first authors and reference ID. The table also contains too much text which makes it difficult to read. ‘Title’ is unnecessary, ‘Sample size’ should be only a number (information on age could be added to target population) and the word ‘low-income’ could be deleted from the ‘Target population’ because in the inclusion criteria you already stated that other populations were excluded from this review. I don’t think that you should include ‘Confounding variables’ and ‘Limitations’ columns because they make Table more complicated. These columns could be included for example in another table (Table 3) with other additional information, or you can mention these factors in the main text. The ‘Results’ should also be a little shorter in the table, avoiding unnecessary words. Type of the study (RCT, cohort, intervention) should be included in the table determining the subgroups (before vs after intervention; intervention vs control group). This is more informative than title of the article. And type of multi-level interventions should be also included (e.g. discount markets AND education). Another relevant data is the time of intervention that is not included in this table.

You might sum up your results in a Figure showing the possible effects and ways of multilevel approach on low-income people.

Some minor comments:

  1. Page 3, row 92: by odds ratio ‘s’ is missing.
  2. Page 3, row 121: NHANES data by Hilary et al: Hilary is the first name of the author, please refer to this work as Seligman et al.
  3. Page 3, row 87-8; Page 3, row 127; page 6 row 274: do not list all authors, only the first author (e.g: Kaur et al; Seligman et al).
  4. Ref Nr 39: authors, title is lacking.
  5. Ref 20 and 23 are the same (Ippolito MM et al, 2017, Public Health Nutr). Please delete the second.
  6. There are a lot of typos in the article, please correct them all (e.g. page 2, row 64, in Table 1, Table 2, page 6 row 241 and 249).

Author Response

Please find the revised manuscript for “Increasing access to healthy foods through improving food environment: a review of mixed methods intervention studies with residents of low-income communities” for possible submission in the Nutrients special issue “Diet Quality, Food Environment and Diet Diversity.” Thank you for taking the time to extensively review our manuscript. We feel that this has given us the opportunity to significantly strengthen the paper, and hope that the revisions meet your expectations.  All changes have included in red in the version with “revision” in the title, and a point by point response has been included below. 

The authors prepared an overview about improving food environment and its role in access to healthy foods in food insecure populations in the USA. It is an interesting, hot topic, but the article in its current form does not seem to be too up to date for readers due to the paucity of current literature (literature from the last two years) and the lack of meta-analyses with the highest level of evidence.

Therefore, during the review process some major and minor comments, questions were raised. Although it is not a systematic review, the authors should at least indicate when the literature was searched. Since the most recent article is from 2020 December, I assume that it was done more than a year ago (early 2021) and that no recent literature search has been done since then. Please clarify it.

Added under Materials and methods, lines 61-62. No papers we considered to meet our criteria were published after December 2020, though in our revision we did add a recent paper on tailoring, lines 368-373.         

You should also clarify the population with age range. Have you studied the effects of food insecurity only in adults, or also in adolescents and children? It should be included in Table 1, Criteria for subject population.

Added in the text on line 68 and table 1. Food insecurity in all age ranges and demographic populations in the United States were considered.

In the recent 2-3 years many systematic reviews and meta-analyses have been published on food insecurity and risk of chronic diseases (obesity, T2DM, CVD), mainly based on data collected in the US population, however I couldn’t find either these meta-analyses or these more recent US (original) publications. Writing on the basis of a single study that food insecurity (FI) predisposes to obesity is less convincing than writing on the basis of a meta-analysis. I know that these meta-analyses use data outside the US also but you can start a statement based on a meta-analysis and then write that US data show also in the same direction. Meta-analysis about FI and obesity: Moradi et al, 2019 (https://pubmed.ncbi.nlm.nih.gov/30219965/), FI and hypertension: Beltrán et al, 2020 (https://pubmed.ncbi.nlm.nih.gov/33201873/),

Thank you. Added data from these papers, lines 125-133 and lines 172-188.

In the T2DM part you cite Seligman et al, 2007 based on data collected 1999-2002, however, more recent data of the same population has also been published (Walker et al, 2018: NHANES 2005-2014). If there is available, why don’t you use the more recent data? This would also confirm the relevance of your publication.

Replaced the study and removed Seligman et al, 2007 completely, lines 155-163.

In part 5 you discuss multilevel intervention studies performed in the US. In rows 198-203 you show results of Hammons et al, however, this publication cannot be found in Table 2. What determined whether a publication was included in this Table or not? This inconsistency is very confusing. All relevant publications should be included.

The title of the table was misleading, we apologize for the confusion on information the table represents and thank you for pointing this out. Table 2 is a representation of the key studies that were mentioned in depth, while the rest of the papers where complementary to these main articles mentioned in Table 2. We have changed the title of the table to better reflect its contents.

I have more concerns about Table 2. It describes multilevel approach studies, however, it is not obvious why are they multilevel? 

A broad explanation of multilevel approaches has been added in the beginning of the paragraph of the multilevel studies on lines 231-237. Additionally, type of approach has been added in the table to indicate levels included in the studies. 

You should include these multiple levels. I don’t think so that Table 2 should contain Title of the articles, this column can be deleted; readers can find these articles in the References based on the first authors and reference ID

Table was adjusted based on this guidance, thank you.

The table also contains too much text which makes it difficult to read. ‘Title’ is unnecessary, ‘Sample size’ should be only a number (information on age could be added to target population) and the word ‘low-income’ could be deleted from the ‘Target population’ because in the inclusion criteria you already stated that other populations were excluded from this review. I don’t think that you should include ‘Confounding variables’ and ‘Limitations’ columns because they make Table more complicated. These columns could be included for example in another table (Table 3) with other additional information, or you can mention these factors in the main text.

Adjustments were made to the table based on these comments, thank you.

The limitations were included in the main text under Discussion and Conclusion, and the confounding variables were taken out since we determined them to not be necessary in the main text and they might  take away the attention from the major points we are focusing on for the paper.

‘Results’ should also be a little shorter in the table, avoiding unnecessary words. Type of the study (RCT, cohort, intervention) should be included in the table determining the subgroups (before vs after intervention; intervention vs control group). This is more informative than title of the article. And type of multi-level interventions should be also included (e.g. discount markets AND education).

Type of study was added as well as the levels under type of approach.

Another relevant data is the time of intervention that is not included in this table.

We had difficulty fitting everything in the table so that it is not too crowded, but appreciate all of your edits. We had a hard time fitting another column for time, but have included it in the narrative when it was not mentioned before on line 240 and line 276.

You might sum up your results in a Figure showing the possible effects and ways of multilevel approach on low-income people.

Thank you, we have added a figure with the different examples of multilevel approaches on the different levels.

Some minor comments:

1. Page 3, row 92: by odds ratio ‘s’ is missing.

Added, line 106.

2. Page 3, row 121: NHANES data by Hilary et al: Hilary is the first name of the author, please refer to this work as Seligman et al.

Changed, line 149.

3. Page 3, row 87-8; Page 3, row 127; page 6 row 274: do not list all authors, only the first author (e.g: Kaur et al; Seligman et al).

Changed, lines 102 and 149.

4. Ref Nr 39: authors, title is lacking.

Added.

5. Ref 20 and 23 are the same (Ippolito MM et al, 2017, Public Health Nutr). Please delete the second.

Deleted second.

6. There are a lot of typos in the article, please correct them all (e.g. page 2, row 64, in Table 1, Table 2, page 6 row 241 and 249).

Corrected, table 1, table 2,  as well as lines 73 and 75. The other typos I believe were part of the section edited for new material.

If any additional changes are required, please let us know. Thank you again.

Reviewer 2 Report

Adopting healthy food can bring co-benefits of human health and planetary health. This review article discussed the impacts of food environment and healthy food accessibility on people's diets in low-income communities. This article is well-written and of significance. I just have a minor comment: the authors claimed that they focus on the residents of low-income communities. Thus, I suggest the authors to 1) describe how the diets of the poor differ from that of the rich; 2) highlight the divergent impacts of food environment on the diets at the different income levels. I suppose that the impact of food environment on diet is very different between the poor and the rich. 

Author Response

Please find the revised manuscript for “Increasing access to healthy foods through improving food environment: a review of mixed methods intervention studies with residents of low-income communities” for possible submission in the Nutrients special issue “Diet Quality, Food Environment and Diet Diversity.” Thank you for taking the time to extensively review our manuscript. We feel that this has given us the opportunity to significantly strengthen the paper, and hope that the revisions meet your expectations.  All changes have included in red in the version with “revision” in the title, and a point by point response has been included below. 

Adopting healthy food can bring co-benefits of human health and planetary health. This review article discussed the impacts of food environment and healthy food accessibility on people's diets in low-income communities. This article is well-written and of significance. I just have a minor comment: the authors claimed that they focus on the residents of low-income communities. Thus, I suggest the authors to 1) describe how the diets of the poor differ from that of the rich; 2) highlight the divergent impacts of food environment on the diets at the different income levels. I suppose that the impact of food environment on diet is very different between the poor and the rich. 

Thank you, this has been added in the introduction on lines 37-38 and included in the paragraph “The impact of food environment on access to healthy food choices”, starting on line 189.

If any additional changes are required, please let us know. Thank you again.

Round 2

Reviewer 1 Report

The authors revised the complete manuscript and answered all my questions. I personally find the new figure very informative. I only have some minor comments:

1.      Page 4, row 125: systematic review instead of systemic.

2.      Page 4 row 137: the direction (negative) of association is missing. This could also mean that higher fruit consumption can increase the mortality from stroke, etc. Please correct it.

3.      Page 4 row 152: I don’t know, why type 2 was deleted, because the section is not discussing Type 1, only type 2 DM.

4.      Please refer to Figure 1 in the main text.

5.      Page 12, Table 2: please include a footage with abbreviations used in the table and their explanation (the table should be interpretable independently of the main text).

6.      Ref 10: please correct to the format used for others.